# From *em*-Projections to Variational Auto-Encoder

**Tian Han**
Department of Computer Science
Stevens Institute of Technology
Hoboken, NJ 07030
than6@stevens.edu

**Jun Zhang**
Department of Psychology
University of Michigan
Ann Arbor, MI 48109
junz@umich.edu

**Ying Nian Wu**
Department of Statistics
University of California, Los Angeles
Los Angeles, CA 90095
ywu@stat.ucla.edu

## Abstract

This paper reviews the *em*-projections in information geometry and the recent understanding of variational auto-encoder, and explains that they share a common formulation as joint minimization of the Kullback-Leibler divergence between two manifolds of probability distributions, and the joint minimization can be implemented by alternating projections or alternating gradient descent.

## 1 Introduction

Generative modeling is a principled framework for unsupervised learning from unlabeled data as well as semi-supervised learning from both unlabeled and labeled data. Taking advantage of the approximation capacity of deep networks [13, 18, 19], deep generative models have achieved impressive successes in unsupervised and semi-supervised learning. A popular deep generative model is the generator model [10, 17], which assumes that the observed example is generated by a latent vector via a top-down network, and the latent vector follows a known prior distribution. The top-down network can be interpreted as a decoder, and it is usually jointly trained with a bottom-up inference network that can be interpreted as an encoder in the framework of variational auto-encoder (VAE) [17].

Generative models can be understood within the framework of information geometry, where each probability distribution is treated as a point, and different families of probability distributions form different manifolds. Information geometry characterizes the geometric structures of manifolds of probability distributions, and has been widely applied to statistics, information science, dynamic systems, etc. [4, 1]. Projections from a probability distribution to a manifold can be formulated as minimizing certain divergence between probability distributions, such as the commonly used Kullback-Leibler (KL) divergence [7].

This paper reviews the information geometric *em*-algorithm which is realized by projections along *e*-geodesic and *m*-geodesic alternatively [8, 2]. We shall also review the recent understanding of variational auto-encoder (VAE) [12]. We explain that *em*-projections and VAE share a common formulation as joint minimization of the Kullback-Leibler divergence between two manifolds of probability distributions, and the joint minimization can be implemented by alternating projections or alternating gradient descent.

---

Deep Learning through Information Geometry workshop at NeurIPS 2020, Vancouver, Canada.

## 2  *em*-algorithm

Consider the probability model $p_\theta(x, z)$ with trainable parameters $\theta$, where $x$ is the observed example, and $z$ is the latent vector. Let $P = \{p_\theta(x, z), \forall \theta\}$ be the manifold formed by the model. We may call $P$ the model manifold.

The information geometric *em*-algorithm considers the class of exponential family model

$$p(x, z; \lambda) = \frac{1}{Z(\lambda)} \exp(\langle \lambda, s(x, z) \rangle), \tag{1}$$

where $s(x, z)$ denotes the sufficient statistics, $\lambda$ denotes the natural parameters, and $Z(\lambda)$ is the normalizing constant.

It is assumed that $p_\theta(x, z) = p(x, z; \lambda(\theta))$ for some parametrization of $\lambda$, where $\lambda(\theta)$ may not be invertible, and the resulting model is a curved exponential family model.

The exponential family model (1) can also be parametrized by its mean parameters $\mu(\lambda) = \mathbb{E}_{p_\lambda}[s(x, z)]$, which is a function of $\lambda$. Assuming $\mu(\lambda)$ is invertible, we can write $p(x, z \mid \mu) = p(x, z; \lambda)$ if $\mu = \mu(\lambda)$. Note here we use the generic notation $p$, but we write $p_\theta$, $p(\cdot; \lambda)$ and $p(\cdot \mid \mu)$ in different styles to distinguish the three different parametrizations.

Define $Q = \{p(x, z|\mu), \mu = s(x_{\mathrm{obs}}, z_{\mathrm{im}}), \forall z_{\mathrm{im}}\}$ where $x_{\mathrm{obs}}$ is fixed at the observed value of $x$, and $z_{\mathrm{im}}$ is the potential imputed value of $z$

The information geometric *em*-algorithm [2, 3] solves the following joint minimization problem:

$$\min_{p \in P, q \in Q} \mathbb{D}_{\mathrm{KL}}(q \| p), \tag{2}$$

where $\mathbb{D}_{\mathrm{KL}}(q \| p) = \mathbb{E}_q[\log(q/p)]$. The algorithm is formulated as follows: initialize at a $p \in P$, and alternate the following two projections (see elementary exposition by [21]):

- *e*-**projection**: $e$-project the current $p$ to $Q$ by minimizing $\mathbb{D}_{\mathrm{KL}}(q \| p)$ over $q \in Q$.
- *m*-**projection**: $m$-project the current $q$ to $P$ by minimizing $\mathbb{D}_{\mathrm{KL}}(q \| p)$ over $p \in P$.

$e$- and $m$-projections derive their names from the manifold of exponential family distributions and the manifold of mixture distributions respectively. The two manifolds have different notions of flatness.

The *em*-algorithm is closely related to the Expectation-Maximization (EM) algorithm [9, 23]. If we define $Q = \{p(x, z|\mu), \mu = \mathbb{E}_{q(z_{\mathrm{im}}|x_{\mathrm{obs}})}[s(x_{\mathrm{obs}}, z_{\mathrm{im}})], \forall q(z|x)\}$, where $q(z|x)$ is an arbitrary imputation distribution, then the *em*-algorithm becomes the EM algorithm.

In the above, we gloss over the difference between a single observation and repeated observations.

See Appendix for an in-depth explanation of information geometry of e- and m-projections.

**Moving beyond exponential family models**. The original *em* and EM algorithms are formulated within the framework of exponential family models (1). In discussing modern deep generative models below, we shall not limit ourselves to exponential family models, and we shall re-define the $Q$ manifold. We shall focus on the key insight of joint minimization (2) and alternating projections. In what follows, we shall connect this insight to the recent understanding of VAE [12].

## 3  VAE as alternating projections

### 3.1  Generator model

The generator model [10, 17] is a non-linear generalization of factor analysis:

$$z \sim \mathrm{N}(0, I_d); \ x = g_\theta(z) + \epsilon, \tag{3}$$

where $x$ is an image, or a sentence, or in general a high-dimensional example, $z$ is a $d$-dimensional latent vector. $I_d$ is the $d$-dimensional identity matrix. $g_\theta(z)$ is a top-down deep network, where $\theta$ consists of all the weight and bias terms. $\epsilon$ is usually assumed to be Gaussian white noise with mean 0 and variance $\sigma^2$. Thus $p_\theta(x|z)$ is $\mathrm{N}(g_\theta(z), \sigma^2 I_D)$, where $D$ is the dimensionality of $x$. The generator model defines the joint probability model or the *complete-data model* $p_\theta(x, z) = p(z)p_\theta(x|z)$, where

$p(z)$ is the known prior model. We define $P = \{p_\theta(x, z), \forall \theta\}$ to be the *model distribution manifold*. The marginal density or the observed-data model is $p_\theta(x) = \int p_\theta(x, z)dz$, which is analytically implicit due to the intractable integral.

Let $p_{\text{data}}(x)$ be the data distribution that generates the observed $x$. The learning of $p_\theta(x)$ can be based on $\min_\theta \mathbb{D}_{\text{KL}}(p_{\text{data}}(x)\|p_\theta(x))$. If we observe training examples $\{x_i, i = 1, ..., n\} \sim p_{\text{data}}(x)$, the above minimization can be approximated by maximizing the log-likelihood

$$L(\theta) = \frac{1}{n}\sum_{i=1}^{n}\log p_\theta(x_i) \doteq \mathbb{E}_{p_{\text{data}}}[\log p_\theta(x)], \tag{4}$$

which leads to maximum likelihood estimate (MLE). In this paper, we assume the sample size $n$ is large enough so that the sample average over $\{x_i, i = 1, ..., n\}$ is essentially the same as expectation with respect to $p_{\text{data}}$.

## 3.2 EM

The EM algorithm can be understood by perturbing the KL-divergence for MLE. Define $D(\theta) = \mathbb{D}_{\text{KL}}(p_{\text{data}}(x)\|p_\theta(x))$. Then $\min_\theta D(\theta)$ is equivalent to $\max_\theta L(\theta)$. Let $\theta_t$ be the estimate at iteration $t$ of EM. Let us consider the following perturbation of $D(\theta)$,

$$\begin{aligned}
S(\theta) &= D(\theta) + \mathbb{D}_{\text{KL}}(p_{\theta_t}(z|x)\|p_\theta(z|x)) \\
&= \mathbb{D}_{\text{KL}}(p_{\text{data}}(x)\|p_\theta(x)) + \mathbb{D}_{\text{KL}}(p_{\theta_t}(z|x)\|p_\theta(z|x)) \\
&= \mathbb{D}_{\text{KL}}(p_{\text{data},\theta_t}(x, z)\|p_\theta(x, z)),
\end{aligned} \tag{5}$$

where we define $p_{\text{data},\theta_t}(x, z) = p_{\text{data}}(x)p_{\theta_t}(z|x)$ as the complete-data distribution at iteration $t$. $S(\theta)$ is a surrogate for $D(\theta)$ at iteration $t$, and $S(\theta)$ is simpler than $D(\theta)$ because $S(\theta)$ is based on the joint distributions of $(x, z)$ instead of the implicit marginal distribution of $x$ as in $D(\theta)$.

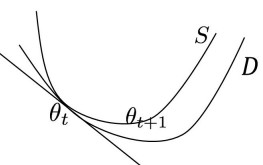

Figure 1: Illustration of EM. The surrogate $S$ majorizes (upper bounds) $D$, and they touch each other at $\theta_t$ with the same tangent.

The perturbation term $\mathbb{D}_{\text{KL}}(p_{\theta_t}(z|x)\|p_\theta(z|x))$, as a function of $\theta$, achieves its minimum 0 at $\theta = \theta_t$. Being a minimum, its derivative at $\theta = \theta_t$ is zero. Thus $S(\theta) \geq D(\theta)$, i.e., $S$ majorizes $D$, and they touch each other at $\theta_t$, where they share the same gradient at $\theta_t$, i.e., $D'(\theta_t) = S'(\theta_t)$. The EM algorithm finds $\theta_{t+1} = \arg\min_\theta S(\theta)$. Since $S(\theta_{t+1}) \leq S(\theta_t)$, and $S$ majorizes $D$, we have $D(\theta_{t+1}) \leq D(\theta_t)$, which is the monotonicity of EM.

The above also underlies the gradient descent algorithm, in particular, the alternating back-propagation algorithm of [11], where $D'(\theta_t)$ is computed by $S'(\theta_t)$.

## 3.3 EM as alternating projections

Define $q(x, z) = p_{\text{data}}(x)q(z|x)$, and re-define $Q = \{q(x, z) = p_{\text{data}}(x)q(z|x).\forall q(z|x)\}$, where $p_{\text{data}}(x)$ is fixed, and $q(z|x)$ can be any conditional distribution of $z$ given $x$. We may call $q(z|x)$ the inference distribution or imputation distribution that imputes the latent $z$ based on $x$. $q(x, z)$ is the *complete-data distribution*. $Q$ is the *data distribution manifold*.

The EM algorithm alternates the following projections:

- **expectation-projection**: Given the current $p$, $\min_q \mathbb{D}_{\text{KL}}(q\|p)$ over $q \in Q$.
- ***m*-projection**: Given the current $q$, $\min_p \mathbb{D}_{\text{KL}}(q\|p)$ over $p \in P$.

The $m$-projection remains the same as in the *em*-projections. The expectation-projection underlies the E-step. In the expectation-projection at iteration $t$ with $\theta_t$, we can write

$$\mathbb{D}_{\mathrm{KL}}(q(x,z)\|p_{\theta_t}(x,z)) = \mathbb{D}_{\mathrm{KL}}(p_{\mathrm{data}}(x)\|p_{\theta_t}(x)) + \mathbb{D}_{\mathrm{KL}}(q(z|x)\|p_{\theta_t}(z|x)), \quad (6)$$

where the conditional KL divergence is defined as

$$\mathbb{D}_{\mathrm{KL}}(q(z|x)\|p(z|x)) = \mathbb{E}_{p_{\mathrm{data}}(x)}\mathbb{E}_{q(z|x)}\left[\log \frac{q(z|x)}{p(z|x)}\right],$$

with the outer expectation with respect to $p_{\mathrm{data}}$. Clearly the minimization in expectation-projection is achieved at $q(z|x) = p_{\theta_t}(z|x)$. The minimization of the $m$-projection is to minimize $S(\theta)$ above.

## 3.4 VAE

The generator model is commonly trained by variational auto-encoder (VAE) [17]. VAE assumes an inference model $q_\phi(z|x)$ as an approximation to the true posterior distribution $p_\theta(z|x)$. In VAE, $q_\phi(z|x)$ is $\mathrm{N}(\mu_\phi(x), V_\phi(x))$, where the mean vector $\mu_\phi$ and the diagonal variance-covariance matrix $V_\phi$ are parametrized by bottom-up inference networks with a new set of parameters $\phi$. For $z \sim q_\phi(z|x)$, we can write $z = \mu_\phi(x) + V_\phi(x)^{1/2}w$, where $w \sim \mathrm{N}(0, I_d)$ is Gaussian white noise. Thus expectation with respect to $z \sim q_\phi(z|x)$ can be written as expectation with respect to $w$. This reparameterization trick [17] helps reduce the variance in Monte Carlo integration. $q_\phi(z|x)$ is the learned inferential computation that approximately samples from $p_\theta(z|x)$. Different from traditional variational inference [16, 20, 6], the parameters $\phi$ in $q_\phi(z|x)$ are shared by all the training examples $x$. $\phi$ and $\theta$ can be trained together by jointly maximizing a lower bound of the log-likelihood.

## 3.5 VAE as alternating projections

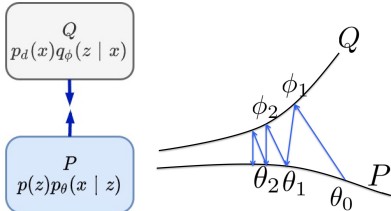

Figure 2: Illustration of variational auto-encoder as alternating projections between model manifold $P$ of complete-data models and data manifold $Q$ of complete-data distributions.

Now we re-define $q_{\mathrm{data},\phi}(x,z) = p_{\mathrm{data}}(x)q_\phi(z|x)$, and re-define $Q = \{q_{\mathrm{data},\phi}(x,z) = p_{\mathrm{data}}(x)q_\phi(z|x), \forall\phi\}$. We can formulate VAE as joint minimization $\min_{p \in P, q \in Q}\mathbb{D}_{\mathrm{KL}}(q\|p)$, where

$$\mathbb{D}_{\mathrm{KL}}(q_{\mathrm{data},\phi}(x,z)\|p_\theta(x,z)) = \mathbb{D}_{\mathrm{KL}}(p_{\mathrm{data}}(x)\|p_\theta(x)) + \mathbb{D}_{\mathrm{KL}}(q_\phi(z|x)\|p_\theta(z|x)), \quad (7)$$

On the right hand side, $D(\theta) = \mathbb{D}_{\mathrm{KL}}(p_{\mathrm{data}}(x)\|p_\theta(x))$ leads to the log-likelihood, and $S(\theta, \phi) = \mathbb{D}_{\mathrm{KL}}(q_{\mathrm{data},\phi}(x,z)\|p_\theta(x,z)) = D(\theta) + \mathbb{D}_{\mathrm{KL}}(q_\phi(z|x)\|p_\theta(z|x))$ is a perturbation of $D(\theta)$. Minimizing $S(\theta, \phi)$ is equivalent to maximizing the evidence lower bound of the log-likelihood.

The minimization can be accomplished by alternating projections:

- **encoder-projection**: Given the current $p$, $\min_q \mathbb{D}_{\mathrm{KL}}(q\|p)$ over $q \in Q$.
- **$m$-projection**: Given the current $q$, $\min_p \mathbb{D}_{\mathrm{KL}}(q\|p)$ over $p \in P$.

Again the $m$-projection remains the same as in the *em*-algorithm. The encoder-projection in VAE is the same as the expectation-projection in EM, except that it is minimization over the inference model or the encoder model $q_\phi(z|x)$, instead of arbitrary $q(z|x)$ as in EM. The optimization is variational inference optimization [16, 20, 6], and the optimized $q_\phi(z|x)$ may have a gap from the true posterior $p_\theta(z|x)$. Figure 2 illustrates the alternating projections. In practice, the alternating projections can be implemented by alternating gradient descent.

The wake-sleep algorithm [15, 14] is similar to VAE, except that it updates $\phi$ by $\min_q \mathbb{D}_{\mathrm{KL}}(P\|Q)$ based on sleep data generated from $P$. Compared to encoder-projection, the order in KL divergence is reversed. As is well known, reversing the order of KL divergence in minimization leads to different behaviors.

### 3.6 Short-run inference

Between EM and VAE, we can strike a middle ground, where we use the short-run MCMC for approximate inference [22]. Specifically, for the true posterior $p_\theta(z|x)$, we can use Langevin dynamics for short-run MCMC:

$$z_0 \sim p(z), \; z_{k+1} = z_k + s\frac{\partial}{\partial z}\log p_\theta(z_k|x) + \sqrt{2s}e_k, \; k = 1, ..., K, \tag{8}$$

where we initialize the dynamics from the fixed prior distribution of $z$, i.e., $p(z) \sim \mathrm{N}(0, I_d)$, and $e_k \sim \mathrm{N}(0, I_d)$ is the Gaussian white noise. The dynamics runs a fixed number of $K$ steps with step size $s$. We use $q_\theta(z|x)$ to represent the distribution of $z_K$. Then the learning with short-run inference is based on

$$\begin{aligned} S(\theta) &= \mathbb{D}_{\mathrm{KL}}(p_{\mathrm{data}}(x)\|p_\theta(x)) + \mathbb{D}_{\mathrm{KL}}(q_{\theta_t}(z|x)\|p_\theta(z|x)) \\ &= \mathbb{D}_{\mathrm{KL}}(p_{\mathrm{data}}(x)q_{\theta_t}(z|x)\|p_\theta(z,x)). \end{aligned} \tag{9}$$

With $\theta_t$ fixed at iteration $t$, $\theta_{t+1}$ is obtained by gradient descent on $S(\theta)$. The learning algorithm is a perturbation of maximum likelihood gradient ascent, where the perturbation is $\mathbb{D}_{\mathrm{KL}}(q_{\theta_t}(z|x)\|p_\theta(z|x))$. As $K \to \infty$, $\mathbb{D}_{\mathrm{KL}}(q_\theta(z|x)\|p_\theta(z|x)) \to 0$ monotonically [7].

Unlike VAE, $q_\theta(z|x)$ does not involve a new set of variational parameters $\phi$. It is still based on the parameters $\theta$ of the generator model. However, we can optimize the small number of algorithmic parameters such as step size by minimizing $\mathbb{D}_{\mathrm{KL}}(q_{\theta_t}(z|x)\|p_\theta(z|x))$ [22].

## 4  Conclusion

This paper reviews the information geometric *em*-algorithm and the recent understanding of VAE. While VAE does not share the framework of exponential family models with the *em*-algorithm, they share the same theoretical framework of joint minimization of KL divergence between two manifolds of distributions, as well as alternating projections (or alternating gradient descent) computation.

See [12] for an information geometric understanding of adversarial learning within the framework of energy-based models, and its integration with variational learning explained in this paper.

## Acknowledgment

The work is supported by NSF DMS-2015577 and AFOSR FA9550-19-1-0213. We thank the reviewers for their insightful comments and suggestions.

## Appendix: information geometry of e- and m-projections

The notion of $e$- and $m$-projections is not restricted to the exponential family, and only relies on the information-geometric notion of $e$- and $m$-flatness of submanifolds that these projections take place. In fact, it is a manifestation of the so-called "generalized Pythagorean relation" [5] satisfied by the Kullback-Leibler divergence $\mathbb{D}_{\mathrm{KL}}(q\|p)$. Let us elaborate below (see Figure 3).

The Kullback-Leibler divergence $\mathbb{D}_{\mathrm{KL}}(\cdot\|\cdot)$ can easily be shown to satisfy the following identity

$$\mathbb{D}_{\mathrm{KL}}(p\|q) + \mathbb{D}_{\mathrm{KL}}(q\|r) - \mathbb{D}_{\mathrm{KL}}(p\|r) = \int (p-q)(\log r - \log q) \tag{10}$$

for any three probability distributions $p, q, r$. Now suppose we construct a one-parameter family $q(t)$, $t \in [0, 1]$, of probability functions connecting $p = q(1)$ to $q = q(0)$ by

$$q(t) = t\,p + (1-t)\,q,$$

and another one-parameter famiy $q(s)$, parameterized by $s \in [0, 1]$, of probability functions connecting $r = q(1)$ to $q = q(0)$ by

$$q(s) = C(s)\,r^s\,q^{1-s}.$$

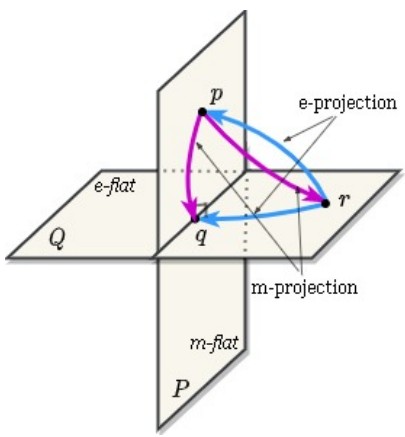

Figure 3: Illustration of $m$-projections from point $p$ (the two red curves) to point $q$ and $r$ in an $e$-flat submanifold $Q$, and $e$-projections from point $r$ (the two blue lines) to point $q$ and $p$ in an $m$-flat submanifold $P$. The curve segments $\widetilde{pq}$ and $\widetilde{qr}$ are orthogonal at point $q$, and hence represent the $m$-projection of $p$ to $Q$ and $e$-projection of $r$ to $P$, respectively. Note that the $e$-geodesic and $m$-geodesic connecting any two points, say, $p$ and $r$, are different.

Here $C(s) = \left( \int r^s q^{1-s} \right)^{-1}$ is the normalization factor, with $C(0) = 1$ and

$$\left. \frac{dC}{ds} \right|_{s=0} = \int q \log \frac{q}{r} = \mathbb{D}_{\mathrm{KL}}(q \| r).$$

Each of these families form a curve segment on the manifold of probability distributions: $\widetilde{pq} := \{q(t)\}_{t \in [0,1]}$ and $\widetilde{qr} := \{q(s)\}_{s \in [0,1]}$ are called $m$-*geodesic* and $e$-*geodesic*, respectively. They meet at the point $q$ where $s = t = 0$. The right-hand side of (10), denoted $\Delta$, becomes

$$\Delta_{(t,s)} = \int \frac{q(t) - q(0)}{t} \left( \frac{\log q(s) - \log q(0)}{s} - \frac{\log C(s) - \log C(0)}{s} \right).$$

Taking $\lim_{t \to 0}, \lim_{s \to 0}$ in the above yields

$$\Delta = \int \frac{dq(t)}{dt} \left( \frac{d \log q(s)}{ds} - \frac{d \log C(s)}{ds} \right) = \int \frac{dq}{dt} \frac{d \log q}{ds} - \frac{d \log C}{ds} \int \frac{dq}{dt} = \int \frac{dq}{dt} \frac{d \log q}{ds},$$

because $\int \frac{dq}{dt} = \frac{d}{dt} \int q = \frac{d}{dt} 1 = 0$. Now suppose we denote $\theta$ as the coordinate system for the manifold of the parametric probability distributions, that is $q = q_\theta$. And the directional derivatives of $\widetilde{pq}$ and $\widetilde{qr}$ (at the intersecting point $q$) are given by, respectively,

$$\frac{d}{dt} = \sum_i V^i \frac{\partial}{\partial \theta^i}, \qquad \frac{d}{ds} = \sum_j U^j \frac{\partial}{\partial \theta^j},$$

where $V$ is the tangent vector for the $m$-geodesic curve $\widetilde{pq}$ at $q$ and $U$ the tangent vector for the $e$-geodesic curve $\widetilde{qr}$ at $q$. Hence

$$\Delta = \sum_{i,j} V^i U^j \int \frac{\partial q_\theta}{\partial \theta^i} \frac{\partial \log q_\theta}{\partial \theta^j} = \sum_{i,j} V^i U^j \int q_\theta \frac{\partial \log q_\theta}{\partial \theta^i} \frac{\partial \log q_\theta}{\partial \theta^j} = \sum_{i,j} V^i U^j g_{ij}(\theta) \equiv \langle V, U \rangle$$

where

$$g_{ij}(\theta) = \int q_\theta \frac{\partial \log q_\theta}{\partial \theta^i} \frac{\partial \log q_\theta}{\partial \theta^j}$$

is the Fisher information matrix, which is the Riemannian (Fisher-Rao) metric on the manifold of the probability functions. If the vectors $V$ and $U$ are chosen in such a way that makes $\Delta = \langle V, U \rangle = 0$, that is, they are "orthogonal" with respect to the Fisher-Rao metric $\langle \cdot, \cdot \rangle$, then (10) becomes the generalized Pythagorean relation

$$\mathbb{D}_{\mathrm{KL}}(p \| q) + \mathbb{D}_{\mathrm{KL}}(q \| r) = \mathbb{D}_{\mathrm{KL}}(p \| r). \tag{11}$$

As each term in (11) is non-negative, we can consider $\mathbb{D}_{\text{KL}}(p\|q)$ as a $m$-projection from $p$ to a submanifold $Q$ formed by the geodesic spray emanating from $q \in Q$, where $q = \operatorname{argmin}_r \mathbb{D}_{\text{KL}}(p\|r)$ is the projection point of the fixed $p$ onto $Q$:

$$\mathbb{D}_{\text{KL}}(p\|q) = \min_{r \in Q} \mathbb{D}_{\text{KL}}(p\|r) \le \mathbb{D}_{\text{KL}}(p\|r).$$

Likewise, we can consider $\mathbb{D}_{\text{KL}}(q\|r)$ as an $e$-projection from $r$ to a submanifold $P$ formed by the geodesic spray emanating from $q \in P$, where $q = \operatorname{argmin}_p \mathbb{D}_{\text{KL}}(p\|r)$ is the projection point of the fixed $r$ onto $P$:

$$\mathbb{D}_{\text{KL}}(q\|r) = \min_{p \in P} \mathbb{D}_{\text{KL}}(p\|r) \le \mathbb{D}_{\text{KL}}(p\|r).$$

The submanifold $P$ is said to be $m$-*flat* and it is made up of a collection of $m$-geodesics of which $\widetilde{pq}$ is an instance, while the submanifold $Q$ is said to be $e$-*flat* and it is made up of a collection of $e$-geodesics of which $\widetilde{qr}$ is an instance. $e$-projection of point $r$ to $P$ (or $m$-projection of point $p$ to Q) is unique when $P$ is $m$-flat (or $Q$ is $e$-flat, respectively). See Figure 4 for an illustration.

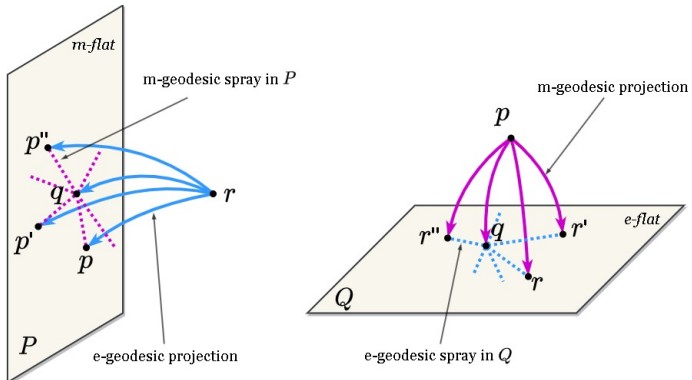

Figure 4: Left: a family of $e$-geodesic projections from a point $r$ (blue filled curves) to an $m$-flat submanifold $P$, along with the $m$-geodesic spray (red dotted lines) emanating from the point $q = \operatorname{argmin}_p \mathbb{D}_{\text{KL}}(p\|r)$, which is the projection point of $r$ onto the $m$-flat submanifold $P$. Right: a family of $m$-geodesic projections from a point $p$ (red filled curves) to an $e$-flat submanifold $Q$, along with the $e$-geodesic spray (blue dotted lines) emanating from the point $q = \operatorname{argmin}_r \mathbb{D}_{\text{KL}}(p\|r)$, which is the projection point of $p$ onto the $e$-flat submanifold $Q$.

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
