# OpenReview forum: "From em-Projections to Variational Auto-Encoder"
_NeurIPS.cc/2020/Workshop/DL-IG — NeurIPSW 2020: DL-IG Oral_

### Official Review · AnonReviewer2 · 2020-10-27
**Interprets the joint minimization of KL divergence in some deep learning models as alternating projections between two manifolds.**

**Rating:** 9
**Confidence:** 4

**Review:**



I enjoyed reading this very well-written paper, and thus recommend its acceptance to the workshop.
The paper interprets the joint minimization of KL divergence in deep learning models as alternating projections between two manifolds.

Here are my comments:

- In Sec 2, it is important to explain when and why the notion of projection amounts to a divergence minimization:
See
https://www.ams.org/journals/notices/201803/rnoti-p321.pdf

- Introduce Fisher metric and orthogonality.
Also important to mention when we have a known uniqueness theorem. (dual connection  projection to autoparallel submanifold)

- Line 37: precise definition of curved exp fams and references would be appreciated.
See for exact conditions of CEFs the book
https://www.springer.com/gp/book/9780387938387


- line 50,  The two manifolds have different notions of flatness.
It is important to say that they are auto-parallel manifolds for the e- and m-connection, and that flatness is related to connections here

- line 54
For modern deep generative models, such a framework is rather limited.
-> please further explain. We are also never too far from an exp fam... (see Efron)

- Bibliography: use {} in .bib to preserve upper cases. Cf [3], [16], etc.

- It would be nice to include the appendix inside the core text.

---

### Official Review · AnonReviewer1 · 2020-10-30
**an EM perspective on VAEs**

**Rating:** 8
**Confidence:** 4

**Review:**

This work offers an EM perspective on VAEs, demonstrating that they can be viewed finding the solution to an alternating minimization procedure akin to the EM algorithm.

I think the work does a good job starting with a sort of purely information geometric result and then progressively relaxing constraints to end up at a modern VAE.  I think the work offers a unique perspective that would help many people better understand and develop a better intuition about what it is that a VAE is attempting to accomplish.

The text is a bit dense at places and could probably benefit from a bit more of a verbose prose.

---

### Decision · Program_Chairs · 2020-11-07

Accept (Oral)